# Weak-Shot Keypoint Estimation via Keyness and Correspondence Transfer

Junjie Chen[1]    Zeyu Luo[1]    Zezheng Liu[1]    Wenhui Jiang[1]    Li Niu[2*]    Yuming Fang[1]

[1]Jiangxi University of Finance and Economics
[2]Shanghai Jiao Tong University
{chenjunjie,2202320619,2202426552,wenhui}@jxufe.edu.cn,
ustcnewly@sjtu.edu.cn, fangyuming@jxufe.edu.cn

## Abstract

Keypoint estimation is a fundamental task in computer vision, but generally requires large-scale annotated data for training. Few-shot and unsupervised keypoint estimation are prevalent economical paradigms, but the former still requires annotations for extensive novel classes while the latter only supports for single class. In this paper, we focus on the task of weak-shot keypoint estimation, where multiple novel classes are learned from unlabeled images with the help of labeled base classes. The key problem is what to transfer from base classes to novel classes, and we propose to transfer keyness and correspondence, which essentially belong to comparing entities and thus are class-agnostic and class-wise transferable. The keyness compares which pixel in the local region is more key, which can guide the keypoints of novel classes to move towards the local maximum (*i.e.*, obtaining precise keypoints). The correspondence compares whether the two pixels belongs to the same semantic part, which can activate the keypoints of novel classes by reinforcing the consistency between two paired images. Extensive experiments and analyses on large-scale benchmark MP-100 demonstrate our effectiveness.

## 1 Introduction

Keypoint estimation is a fundamental computer vision task and has extensive applications in real world, including intelligent interaction [65], behavioural analysis [2] and augmented reality [46]. Although existing keypoint estimation methods [1, 5, 53] have achieved great success, they usually require large-scale annotated data of all classes for fully supervised learning. As a consequence, the expensive annotation dramatically limits category-wise expansion and wider application.

Few-shot learning and unsupervised learning are two prevalent paradigms to economize annotations. As illustrated in Fig. 1 (a), few-shot keypoint estimation [69, 59, 9] greatly reduces the number of labeled images of novel classes (*i.e.*, novel object categories), and unsupervised keypoint estimation [64, 79, 17] reduces the annotations of target class. However, few-shot methods still requires non-negligible annotations for each novel class, and thus the annotation cost can become substantially high with a large number of novel classes. Unsupervised methods only focuses on single class, and is difficult to promise that the discovered keypoints are desired. Thus, it would be more practicable if we can learn desired keypoints from unlabeled images for multiple novel classes.

In this paper, we follow the paradigm of weak-shot learning [7, 38, 8, 26] and explore the task of weak-shot keypoint estimation. Weak-shot learning has achieved promising success in economizing annotations for various tasks, including classification [7], detection [38, 80, 32], segmentation [8, 22, 26, 3, 81], and so on [61]. Specifically in keypoint estimation, we would like to learn keypoints

---

*Corresponding author

39th Conference on Neural Information Processing Systems (NeurIPS 2025).

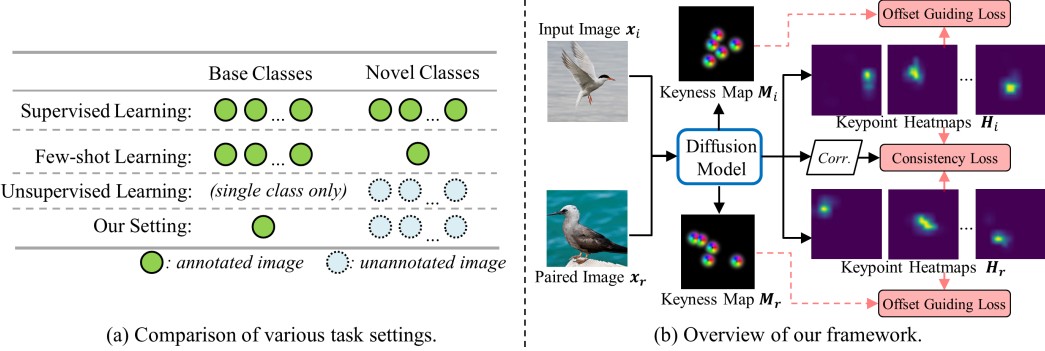

(a) Comparison of various task settings.     (b) Overview of our framework.

Figure 1: (a) Data comparison among fully supervised learning, few-shot learning, unsupervised learning and our weak-shot learning. Our setting is more economical and practicable. (b) Our proposed framework transferring keyness and correspondence from base classes to support the unsupervised learning of multiple novel classes with offset guiding loss and consistency loss.

for novel classes from unlabeled images with the support of labeled images of base classes, as shown in Fig. 1 (a). Intuitively, we leverage the knowledge transferred from base classes to facilitate the unsupervised learning of multiple novel classes. Therefore, our setting integrates few-shot and unsupervised keypoint estimation to alleviate their drawbacks (*i.e.*, annotations on novel classes, single-class discovery, and uncontrollable discovery).

In weak-shot keypoint estimation, the key problem is what to transfer to facilitate the unsupervised learning of novel classes. A representative paradigm [63, 7] in previous works is to transfer the comparing entities (or relation), which is class-agnostic and class-wise transferable. Different from the image-level comparison in [63, 7], here we explore to compare pixels to facilitate keypoint localization. Specifically, we conduct the following two types of comparisons: intra-image comparison (*i.e.*, keyness) and inter-image comparison (*i.e.*, correspondence).

Respectively, for intra-image comparison, we propose to compare the pixels in local regions, since "keypoint" can be understood as the most key point within a local region or the point that are more key compared to nearby ones (dubbed as "keyness"). Given a keypoint Gaussian map, we compute the two-dimensional derivative as its keyness map, where each pixel is a 2-D vector indicating the more key direction. We learn such keyness from the annotations of base classes, and employ the transferred keyness to the keypoints of novel classes to move towards the local maximum (*i.e.*, by offset guiding loss). For inter-image comparison, we propose to compare the pixels in two images belonging to the same class, since "keypoint" should be semantic consistency across images. Given two images from the same class, we compute the bipartite pairwise similarities to derive correspondences as in [44]. Analogous to keyness transfer, we class-wise transfer correspondences to activate the keypoints of novel classes (*i.e.*, by consistency loss).

Based on aforementioned keyness and correspondence transfer, we design a well-tailored and effective model for weak-shot keypoint estimation, as shown in Fig. 1 (b). In particular, we follow [18] and build our model upon pretrained diffusion model [55], because unsupervised keypoint estimation is quite challenge, let alone in the multi-class scenario. The training data contains image pairs from both base classes and novel classes. For the image pair from base classes, we enable the loss terms to learn keyness and correspondence. Otherwise, we switch the loss terms and apply estimated keyness and correspondence to support the unsupervised learning of novel classes.

We conduct comprehensive experiments and in-depth analyses on the large-scale multi-category pose dataset MP-100 [69]. Because the model has never accessed the GT keypoints of novel classes, we adopt the matching-based evaluation [13] and regression-based evaluation [18] for quantitative analysis. Our contributions could be summarized as follows:

(1) We are the first to explore weak-shot keypoint estimation, where we could use the knowledge transferred from base classes to facilitate the unsupervised learning of multiple novel classes.

(2) We propose a well-tailored framework to learn keyness and correspondence from base classes, and transfer them to provide effective supervision for the unsupervised learning of novel classes.

(3) Extensive experiments on the large-scale dataset demonstrate the effectiveness of our method.

## 2 Related Works

**Fully-Supervised Pose Estimation.** Most existing methods focus on fully-supervised learning for specific class, including human [1, 29], animals [5, 27], and vehicles [53, 60]. Technically, these methods could be roughly classified into regression-based [49, 28, 35, 36, 15], heatmap-based [75, 10, 20, 21, 24, 47], and transformer-based [45, 70, 33, 58, 74] methods. For example, DEKR [15] employed adaptive convolutions through spatial transformer to activate keypoint regions and learn representations for more accurate keypoints. Above works have achieved great success for estimating pose of specific classes, but they require abundant annotations and are inapplicable for novel classes. In this paper, we focus on estimating poses for multiple novel classes.

**Zero-Shot and Few-Shot Keypoint Estimation.** Zero-shot keypoint estimation locates keypoints for novel classes using descriptions, and existing methods generally learn a mapping from description to keypoints [71, 72, 56, 42, 76, 77, 78]. Our task setting is closer to few-shot keypoint estimation, which locates keypoints for novel classes using a few examples. Early methods focus on specific domains, *e.g.*, facial images [4, 67], clothing images [14], or animal images [62]. Recently, POMNet [69] introduced the task of category-agnostic keypoint estimation, and proposed a matching framework to retrieve results based on few-shot. Later, CapeFormer [59] further enhanced the similarity modeling and proposed to refine the coarse keypoints via a transformer decoder. Meanwhile, Lu *et al.* [40] explored a more flexible few-shot scenario to learn novel/base classes and novel/base keypoints. Although existing methods [43, 41, 19, 48, 30, 31, 54, 52] have great promoted few-shot keypoint estimation, they requires non-negligible annotations for novel classes. In contrast, we propose to harness diffusion models and learn novel classes from unannotated images.

**Unsupervised Keypoint Estimation.** Unsupervised keypoint estimation only require unlabeled images to discovery keypoints for single class. Overall, the high-level idea of most existing method is to design keypoint bottleneck in image reconstruction [64, 17, 16, 39] or video reconstruction [23, 57]. Recently, Hedlin *et al.* [18] proposed to employ attention maps of text embeddings in pretrained diffusion models as keypoint heatmaps, and optimize text embeddings with localization loss and equivariance loss for unsupervised keypoint estimation. Although above methods could discover keypoints from unlabeled images, most of them are difficult to discover desired keypoint for multiple classes. In this work, we take the inspiration of [18] and propose a well-tailored framework to transfer keypoint prompts and correspondences for weak-shot keypoint estimation.

**Weak-Shot Learning** Deep learning methods generally require large-scale labeled data, and thus the demand of reducing annotations is extensive in various tasks. To this end, weak-shot learning, *i.e.*, learning weakly labeled novel classes with the support of fully labeled base classes, has been explored in classification [7, 51, 50], detection [38, 80], semantic segmentation [8, 81], instance segmentation [22, 26], and so on [61]. To name a few, SimTrans [7] proposed to transfer the pair-wise similarity to de-noise the web data of novel classes. RETAB [81] proposed to transfer affinity and boundary to expand seed mask to semantic mask. Here we explore weak-shot keypoint estimation, and propose to transfer keyness and correspondence to support multiple novel classes.

## 3 Method

### 3.1 Task Setting of Weak-Shot Keypoint Estimation

In weak-shot keypoint estimation, we would like to learn keypoints for novel classes from unlabeled images with the support of labeled images of base classes. Specifically, in the training stage, there are labeled samples from base classes available, and the $i$-th labeled sample consists of image $x_i \in \mathbb{R}^{3 \times H_{full} \times W_{full}}$ and GT keypoints $P_i^* \in \mathbb{R}^{K_i^* \times 2}$. Note that, $K_i^*$ is the keypoint number, and the images from different classes could have different keypoint numbers. For ease of process, we pad all keypoints to a unified number $K$ with visibilities, *i.e.*, $[P_i^*; V_i^*] \in \mathbb{R}^{K \times (2+1)}$. There are also unlabeled samples for novel classes available, which have no overlap with base classes. In the test stage, the model should locate the keypoints for images from multiple novel classes.

### 3.2 The Forward Pipeline of Our Framework

As shown in Fig. 2, we propose a compact framework for learning and transferring keyness and correspondence. In the training stage, each mini-batch contains images randomly sampled from

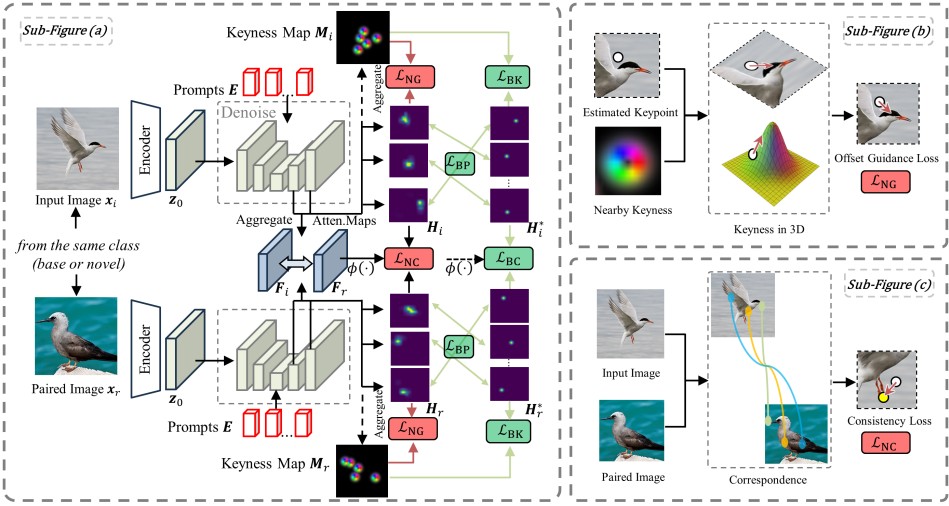

Figure 2: (a) The detailed illustration of our framework in the training stage. Given image pairs from the same base or novel class, we employ keypoint prompts to estimate keypoint heatmaps and aggregate feature maps to estimate keyness and extract correspondences. For labeled image pairs from base classes, we learn keypoint prompts, keyness and correspondence via $\mathcal{L}_{\text{BP}}$, $\mathcal{L}_{\text{BK}}$ and $\mathcal{L}_{\text{BC}}$. For unlabeled image pairs from novel classes, we transfer them to learn valid keypoints via $\mathcal{L}_{\text{NG}}$, $\mathcal{L}_{\text{NC}}$ and $\mathcal{L}_{\text{NU}}$. The sub-figures (b) and (c) illustrate the insight of $\mathcal{L}_{\text{NG}}$ and $\mathcal{L}_{\text{NC}}$.

both base classes and novel classes. Additionally, we pair each training image $\boldsymbol{x}_i$ with a reference image $\boldsymbol{x}_r$ (i.e., paired image) from the same class, and apply respective supervisions on image pairs from base or novel classes. To estimate keypoints, we follow [18] and maintain $N$ learnable keypoint prompts $\boldsymbol{E} \in \mathbb{R}^{N \times D}$, which correspond to text/query embeddings and set as $N = 200$ by default. Different from the single-class scenario in [18], our keypoint prompts are shared by all images and all classes, and thus could learn transferrable knowledge under our match-based supervision.

Firstly, for each image in the input pair, we obtain keypoint heatmaps according to keypoint prompts $\boldsymbol{E}$. We denote the $\Phi_l^c(\cdot)$ and $\Psi_l^c(\cdot)$ as the $c$-th head and the $l$-th linear layers of the U-Net in the transformer part of pretrained Diffusion model. We compute the query as $\boldsymbol{Q}_l^c = \Phi_l^c(\boldsymbol{z}_{t=1}) \in \mathbb{R}^{H \times W \times D_l}$, where $\boldsymbol{z}$ is the latent embedding mapped from image $\boldsymbol{x}$, and we use $t = 1$ as suggested by [18]. We compute the key from keypoint prompts $\boldsymbol{K}_l^c = \Psi_l^c(\boldsymbol{E}) \in \mathbb{R}^{N \times D_l}$. Then we obtain keypoint heatmaps $\boldsymbol{H} \in \mathbb{R}^{H \times W \times N}$ by collecting cross-attention maps from various layers:

$$\boldsymbol{H} = \mathbb{E}_{l=7..10}\left[\mathbb{E}_c\left[\text{softmax}_c(\boldsymbol{Q}_l^c \cdot \boldsymbol{K}_l^c / \sqrt{D_l})\right]\right]. \tag{1}$$

In this way, we can obtain the estimated keypoint heatmaps $\boldsymbol{H}_i \in \mathbb{R}^{H \times W \times N}$ and $\boldsymbol{H}_r \in \mathbb{R}^{H \times W \times N}$ for images $\boldsymbol{x}_i$ and $\boldsymbol{x}_r$ using keypoint prompts $\boldsymbol{E}$. More details could be found in [18].

Secondly, we aggregate the feature maps from above diffusion process and obtain hyperfeatures $\boldsymbol{F} \in \mathbb{R}^{H \times W \times D}$ for estimating keyness. Concretely, $\boldsymbol{F} = \sum_s^S \sum_l^L w_{l,s} \cdot f_l(F_{l,s})$, where $f_l(F_{l,s})$ is squeezed feature map, $S$ is the number of subsampled timesteps selected from the diffusion timesteps, and $w_{l,s}$ denotes the mixing weights for the $l$-th layer at the $i$-th timestep, and we use the architecture of [44]. Since keyness compares the pixels in local regions, we intuitively estimate it via conv. layers with act. functions. We denote the estimated keyness map as $\boldsymbol{M} \in \mathbb{R}^{2 \times H \times W}$, where each pixel $\boldsymbol{M}[x, y]$ is a 2D vector indicating the direction to the most key pixel.

Finally, we use the hyperfeatures of the paired images to compute the semantic correspondences, denoted as $\boldsymbol{F}_i$ and $\boldsymbol{F}_r$ respectively for images $\boldsymbol{x}_i$ and $\boldsymbol{x}_r$. Specifically, we compute the mapping $\phi(\cdot)$ from keypoint to its corresponding keypoint in the other image as:

$$\phi(\boldsymbol{p}_i)_{\rightarrow r} = \arg\max_{\boldsymbol{p}_r} S(\boldsymbol{F}_i[\boldsymbol{p}_i], \boldsymbol{F}_r[\boldsymbol{p}_r]), \tag{2}$$

where $\boldsymbol{p}_i$ and $\boldsymbol{p}_r$ are points on two paired images and $S(\cdot, \cdot)$ denotes cosine similarity. As in [44], we only need to compute the similarity between all keypoints on $\boldsymbol{x}_i$ and all pixels on $\boldsymbol{x}_r$, and thus the computation is relatively efficient and affordable.

Given above estimated keypoint heatmaps $(\boldsymbol{H}_i, \boldsymbol{H}_r)$, keyness maps $(\boldsymbol{M}_i, \boldsymbol{M}_r)$, and correspondence $\phi(\cdot)$ for image pair $(\boldsymbol{x}_i, \boldsymbol{x}_r)$, we switch the loss terms for learning from base classes and transferring to novel classes. For labeled image pairs from base classes, we learn keypoint prompts, keyness and correspondence via $\mathcal{L}_{\mathrm{BP}}$, $\mathcal{L}_{\mathrm{BK}}$ and $\mathcal{L}_{\mathrm{BC}}$. For unlabeled image pairs from novel classes, we transfer them to learn valid keypoints via $\mathcal{L}_{\mathrm{NG}}$, $\mathcal{L}_{\mathrm{NC}}$ and $\mathcal{L}_{\mathrm{NU}}$. We introduce the loss terms as following.

### 3.3 Learning from Base Classes

To learn keypoint prompts from base classes, we apply following loss over the keypoint heatmaps:

$$\mathcal{L}_{\mathrm{BP}} = \frac{1}{K} \sum_k^K \left\| \boldsymbol{H}_i[\delta(k)] - \boldsymbol{H}_i^*[k] \right\|^2 + \left\| \boldsymbol{H}_r[\delta(k)] - \boldsymbol{H}_r^*[k] \right\|^2, \quad (3)$$

where $\boldsymbol{H}_i^*[k]$ is the $k$-th GT keypoint Gaussian map generated as in [69]. Besides, the max value for $\boldsymbol{H}_i^*[k]$ is 1 if the $i$-th keypoint is visible, otherwise 0. $\delta(\cdot)$ is the matching function solved as in [6, 11, 9]. Although it is feasible to directly apply the matching-loss to each image respectively as in [9, 11, 6], we propose to conduct joint matching over two paired images in light of the semantic consistency of keypoints. Specifically, we calculate the cost $\boldsymbol{C} \in \mathbb{R}^{N \times K}$ by considering the cost over paired two images *i.e.*, $\boldsymbol{C}[n,k] = \|\boldsymbol{H}_i[n] - \boldsymbol{H}_i^*[k]\|^2 + \|\boldsymbol{H}_r[n] - \boldsymbol{H}_r^*[k]\|^2$. Such loss can adaptively embed knowledge into the same matched prompt, leading to transferability.

To learn valid keyness, we apply the loss analogous to Eqn. 3 due to the same "map" representation:

$$\mathcal{L}_{\mathrm{BK}} = \left\| \boldsymbol{M}_i - \boldsymbol{M}_i^* \right\|^2 + \left\| \boldsymbol{M}_r - \boldsymbol{M}_r^* \right\|^2, \quad (4)$$

where $\boldsymbol{M}^* \in \mathbb{R}^{2 \times H \times W}$ indicate the GT keyness map of input image or paired image, where each pixel $\boldsymbol{M}[x,y]$ is a 2D vector indicating the direction to the most key pixels. We compute the derivative of GT keypoint Gaussian map to obtain GT keyness map, *i.e.*, the GT of $\boldsymbol{M}[x,y]$ is the derivative of the GT keypoint Gaussian map at $[x,y]$. To learn valid correspondences, we compute cosine similarity between every possible pair of points and apply a symmetric cross entropy loss according to GT keypoints as [44], which is denoted as $\mathcal{L}_{\mathrm{BC}}$.

Note that, the names of above loss terms (*i.e.*, $\mathcal{L}_{\mathrm{BP}}$, $\mathcal{L}_{\mathrm{BK}}$ and $\mathcal{L}_{\mathrm{BC}}$) all begin with **B**, indicating that they are enabled only when the image pair comes from **B**ase classes.

### 3.4 Transferring to Novel Classes

For the unlabeled image pair $\boldsymbol{x}_i$ and $\boldsymbol{x}_r$ belonging to novel classes, we transfer and reuse the keypoint prompts $\boldsymbol{E}$ to estimate keypoint heatmaps $\boldsymbol{H}_i$ and $\boldsymbol{H}_r$. Note that such estimations without accessing any data of novel classes satisfyingly locate keypoints for multiple novel classes, which will be demonstrated in Sec. 4.3. Nevertheless, the domain gap between base classes and novel classes inevitably matters. To bridge the domain gap, we propose to apply offset guiding loss and consistency loss according to the transferred keyness and correspondences.

Specifically, we firstly convert the heatmaps $\boldsymbol{H}_i$ and $\boldsymbol{H}_r$ to the keypoint coordinates and visibilities $[\boldsymbol{P}_i; \boldsymbol{V}_i]$ and $[\boldsymbol{P}_r; \boldsymbol{V}_r] \in \mathbb{R}^{N \times (2+1)}$ via the soft argmax and max operators in [59]. Given the transferred keyness map, we guide the coordinates with following loss:

$$\mathcal{L}_{\mathrm{NG}} = \frac{1}{N} \sum_k^N \left\| \boldsymbol{P}_i[k] - dt(\boldsymbol{P}_i[k] + \boldsymbol{M}_i[\boldsymbol{P}_i[k]]) \right\|^2 + \left\| \boldsymbol{P}_r[k] - dt(\boldsymbol{P}_r[k] + \boldsymbol{M}_r[\boldsymbol{P}_r[k]]) \right\|^2, \quad (5)$$

where $\boldsymbol{M}_i[\boldsymbol{P}_i[k]]$ is an offset vector indicating the direction for $\boldsymbol{P}_i[k]$ to be more key, and $dt(\cdot)$ detaches and blocks the gradient to provide stable supervision. By such offset guiding loss, the estimated keypoints could be supervised to gradually offset to more key positions.

For the consistency loss, we reuse the aggregation network to obtain pseudo-points $\widetilde{\boldsymbol{P}}_r \in \mathbb{R}^{N \times 2}$ by corresponding $\boldsymbol{P}_i$ from image $\boldsymbol{x}_i$ to image $\boldsymbol{x}_r$, *i.e.*, $\widetilde{\boldsymbol{P}}_r = \phi(\boldsymbol{P}_i)_{\rightarrow r}$. We obtain pseudo-points $\widetilde{\boldsymbol{P}}_i$ similarly. Then, we use below consistency loss to facilitate the keypoint learning of novel classes:

$$\mathcal{L}_{\mathrm{NC}} = \sum_n^N \boldsymbol{V}_r[n] \cdot \left\| \boldsymbol{H}_i[n] - \mathcal{H}(\widetilde{\boldsymbol{P}}_i[n]) \right\|^2 + \boldsymbol{V}_i[n] \cdot \left\| \boldsymbol{H}_r[n] - \mathcal{H}(\widetilde{\boldsymbol{P}}_r[n]) \right\|^2, \quad (6)$$

where $\boldsymbol{V}_r[n]$ is the visibility of the $n$-th pseudo-point blocking wrong supervision from invisible pseudo-point, and $\mathcal{H}(\cdot)$ is a function mapping keypoint coordinates to Gaussian heatmap in [69]. By our consistency loss, the keypoint heatmaps estimated on the unlabeled images of novel classes are regularized to approximate to the corresponding keypoints on the paired images.

Besides, we apply the equivariance and localization loss to complement our unsupervised learning on novel classes, which are proposed in [18] for single-class unsupervised learning. We use the same configuration and denote as $\mathcal{L}_{\mathrm{NU}}$. Note that, the names of above losses (*i.e.*, $\mathcal{L}_{\mathrm{NG}}$, $\mathcal{L}_{\mathrm{NC}}$ and $\mathcal{L}_{\mathrm{NU}}$) all begin with **N**, indicating that they are enabled when the image pair comes from **N**ovel classes.

### 3.5 Summary of Training and Inference

Our framework is end-to-end learnable, which can jointly learn keypoint prompts, keyness and correspondence from base classes and transfer them to novel classes. In the training stage, our full loss for the $i$-th image $\boldsymbol{x}_i$ can be summarized as:

$$\mathcal{L}_{\mathrm{FULL}} = \mathbf{1}_{\boldsymbol{x}_i \in \mathcal{B}} \cdot (\mathcal{L}_{\mathrm{BP}} + \mathcal{L}_{\mathrm{BK}} + \alpha \cdot \mathcal{L}_{\mathrm{BC}}) + \mathbf{1}_{\boldsymbol{x}_i \in \mathcal{N}} \cdot (\beta \cdot \mathcal{L}_{\mathrm{NG}} + \gamma \cdot \mathcal{L}_{\mathrm{NC}} + \lambda \cdot \mathcal{L}_{\mathrm{NU}}). \quad (7)$$

where $\mathbf{1}_{(\cdot)}$ is the indicator function, $\mathcal{B}$ (*resp.*, $\mathcal{N}$) denotes the image set of base classes (*resp.*, novel classes). Hence, we respectively apply the supervisions for the base images and novel images. As for the loss balancing, we find that $\mathcal{L}_{\mathrm{BP}}$ balance well with $\mathcal{L}_{\mathrm{BK}}$, may due to the same map representation. Besides, $\alpha = 0.1$, $\beta = 0.2$, $\gamma = 0.1$ and $\lambda = 0.5$ are hyper-parameters for balancing the other losses.

In the test stage, we remove the additional modules estimating keyness and correspondence, which are only used to provide effective supervisions for learning novel classes without annotations. Compared with the related work [18] in unsupervised keypoint estimation, our model keeps the same architecture for inference and has extra capacity to estimate keypoints for multiple unlabeled classes.

## 4 Experiments

### 4.1 Dataset and Implementation Details

We follow the related works [69, 59, 9] to conduct our experiments on MP-100 dataset [69], which is the most prevalent benchmark dataset and covers 100 classes within 8 super-classes. MP-100 dataset contains keypoint numbers ranging from 8 to 68 across different classes. We follow the dataset splits in [69], *i.e.*, all classes are split into non-overlapping train/val/test sets with the ratio of $70 : 10 : 20$, and there are five random splits (S1-S5) to reduce the impact of randomness. Considering that our model is built on pretrained diffusion model, we focus on learning from a small-scale training set in this paper, similar to large model adaption [66]. Unless otherwise stated, we use 3 labeled images per base class and 30 unlabeled images per novel class by default.

In practice, the keypoint orders of different classes matter transfer learning. Specifically, the default orders in MP-100 are related, *e.g.*, the first keypoint of base/novel classes always corresponds to "left-eye". On the one hand, such relation requires that all class annotations keep explicit relation and thus severely limits class-wise extension. On the other hand, models may class-wise transfer by overfitting such unpractical relation. Therefore, we use the shuffled orders of each class in experiments, which has less limitation and thus more practical.

### 4.2 Evaluation and Metric

Similar to unsupervised keypoint estimation [18], our model have never accessed the GT annotations of novel classes, and thus we cannot directly compare the estimated keypoints with GT keypoints. Thus, we employ the following two kinds of evaluations.

**Matching-based evaluation** Considering that the number of GT keypoints varies across classes, we follow unsupervised semantic segmentation [13] to compare the estimation with GT based on matching. We employ Average Precision (AP) to compute the metric, because unmatched keypoints will reduce recall rate and thus reduce AP. Specifically, for the test image $\boldsymbol{x}_i$ with padded GT keypoints $[\boldsymbol{P}_i^*; \boldsymbol{V}_i^*] \in \mathbb{R}^{K \times (2+1)}$ come from novel class, the model produces padded keypoints $[\boldsymbol{P}_i; \boldsymbol{V}_i] \in \mathbb{R}^{N \times (2+1)}$. Then, we compute the matching cost over all test images in each class by:

$$\mathcal{C}_{n,k} = \sum_i \boldsymbol{V}_i^*[k] \cdot \mathbf{1}_{(|\boldsymbol{P}_i[n] - \boldsymbol{P}_i^*[k]| < T_p)} \cdot \mathbf{1}_{(\boldsymbol{V}_i[n] > T_v)}, \quad (8)$$

Table 1: Comparison with prior works. We report results on 5 dataset splits with matching-based and regression-based evaluation.

| Method | MP-100 Split1 | | MP-100 Split2 | | MP-100 Split3 | | MP-100 Split4 | | MP-100 Split5 | | AVG | |
|---|---|---|---|---|---|---|---|---|---|---|---|---|
| | mAP↑ | L2↓ | mAP↑ | L2↓ | mAP↑ | L2↓ | mAP↑ | L2↓ | mAP↑ | L2↓ | mAP↑ | L2↓ |
| Sim.Base. [68] | 14.4 | 60.8 | 12.9 | 65.3 | 13.3 | 63.5 | 13.1 | 64.4 | 14.2 | 61.3 | 13.6 | 63.1 |
| GroupPose [37] | 17.4 | 55.7 | 16.7 | 58.2 | 17.0 | 57.1 | 15.9 | 58.4 | 18.2 | 52.1 | 17.0 | 56.3 |
| He *et al.* [16] | 18.3 | 51.5 | 16.2 | 56.6 | 18.6 | 51.7 | 17.8 | 54.3 | 19.5 | 50.2 | 18.1 | 52.9 |
| Hedlin *et al.* [18] | 27.0 | 38.3 | 22.1 | 50.3 | 23.5 | 38.6 | 22.1 | 44.2 | 24.1 | 47.1 | 23.8 | 43.7 |
| MetaPoint [9] | 31.5 | 33.5 | 36.2 | 31.7 | 30.1 | 22.4 | 26.8 | 25.9 | 36.9 | 39.6 | 32.3 | 30.6 |
| Ours | **70.4** | **18.4** | **61.3** | **26.2** | **60.7** | **19.6** | **61.7** | **21.3** | **58.9** | **24.3** | **62.6** | **22.0** |
| Oracle* | 86.1 | 14.4 | 78.3 | 21.6 | 72.8 | 16.9 | 69.6 | 19.4 | 76.0 | 20.3 | 76.6 | 18.5 |

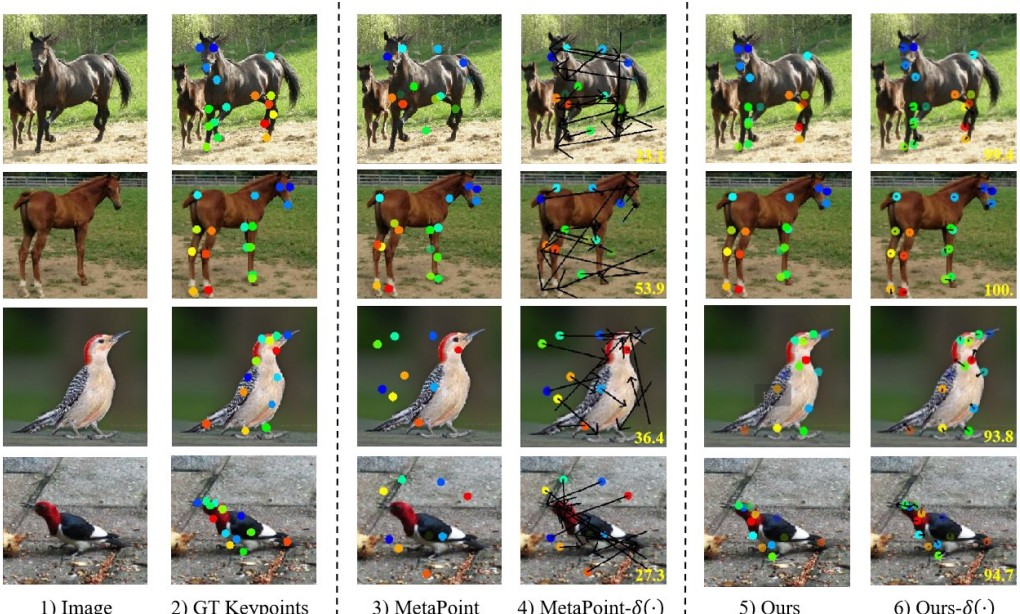

| 1) Image | 2) GT Keypoints | 3) MetaPoint | 4) MetaPoint-$\delta(\cdot)$ | 5) Ours | 6) Ours-$\delta(\cdot)$ |

Figure 3: The qualitative comparison against the most competitive baseline. The first two columns show the image and GT keypoints. The mid two columns display the localized keypoints and matched keypoints of MetaPoint[9]. Note that we use transparency to show the visibility, the right-bottom number indicates the F1-score of keypoint of this sample, and the black arrows show the deviations to matched GT. The right two columns display the same results of our method.

where $\mathbf{1}_{(\cdot)}$ is the indicator function, $T_p$ is a distance threshold as in PCK [73] and $T_v$ is a score threshold for computing AP. We follow [69] to adopt $T_p = 0.2$, and use 9 scores from 0.1 to 0.9 to calculate AP. Finally, we obtain our overall metric by averaging the AP of all classes, *i.e.*, mAP↑.

**Regression-based evaluation** We also follow related work [18, 17, 79, 23, 39] to compare estimation with GT based on linear regression for better comprehensiveness. Specifically, a linear regression model is firstly trained to map estimated keypoints to padded GT keypoints on val images. Afterwards, in the formal evaluation, all the estimated keypoints on test images are mapped by above regressor, and the averaged L2↓ error to GT is employed as overall metric.

## 4.3  Comparison with Prior Works

**Comparable baselines.** As far as we know, there is no method designed for weak-shot keypoint estimation, and thus we adapt representative methods to our baselines, including Sim.Base. [68], GroupPose [37], He *et al.* [16], Hedlin *et al.* [18] and MetaPoint [9]. Specifically, Sim.Base. [68], He *et al.* [16], and Hedlin *et al.* [18] estimate the heatmaps of padded GT keypoints directly. GroupPose [37] regresses the coordinates of padded GT keypoints directly. MetaPoint [9] contains the first stage in [9], which learn potential keypoints with matching-based supervision. We supervise

all methods with heatmap loss or coordinates loss on labeled images of base classes, and complement all methods with unsupervised loss (*i.e.*, equivariance loss and localization loss as in [18]) on unlabeled images of novel classes. We also set a baseline named Oracle* to show our upper-bound by training our model with all labeled images from both base class and novel classes.

**Quantitative comparison.** We summarize the results of above baselines and our method on five dataset splits in Tab. 1. Firstly, we could find that our method outperforms baselines by large margins in both evaluation metrics. Although Hedlin *et al.* [18] could utilize pretrained diffusion model, it cannot adaptively learn transferrable keypoint prompts and thus only achieves dissatisfactory performances. MetaPoint [9] can learn transferrable keypoints, but employs common matching-based loss on single image and also requires abundant annotated images for training. By contrast, our model can leverage pretrained diffusion model to adaptively learn and transfer keypoint prompts and correspondences, resulting in preferable performances. Besides, our model can reach about 75% of the upper-bound mAP represented by Oracle*, showing promising potential. Overall, our proposed model has relatively robust mAPs on different splits using both evaluation protocols.

**Qualitative comparison.** We select the most competitive baseline (*i.e.*, MetaPoint [9]) for qualitative comparison in Fig. 3. With threshold $T_v = 0.5$, we show the F1-score (harmonic mean of precision and recall) of matched keypoints in the right-bottom of sub-figures. In the top two rows, the baseline may locate right keypoints for horse, but fails to keep semantic consistency across images (by comparing two rows). In the bottom two rows, the baseline seems to fail to locate keypoints for bird, may due to the challenging domain gap between base and novel classes. Overall, the keypoints located by our model could better fit the object structures and keep semantic consistency across images, which is a general requirement in keypoint estimation. Therefore, our method could learn preferable keypoints in weak-shot keypoint estimation.

## 4.4 Comparison with Other Settings

Our weak-shot keypoint estimation is related with two prevalent settings (*i.e.*, few-shot and unsupervised setting) as shown in Fig. 1. Here we conduct setting-wise comparison to investigate our potential.

**Comparison with few-shot setting.** We set representative few-shot baselines including PoseAnything [19], MetaPoint [9] and SCAPE [34]. Considering that Peng *et al.* [52] requires test-time optimization, we don't include it as baseline. As shown in Tab. 2, few-shot methods [19, 9, 34] generally achieve inferior performances in our experiments, due to the limited training data (3 labeled images per base class described in Sec. 4.1). Although few-shot methods may outperform our method by using extra labeled support images from novel classes, the support images are not always available for all novel classes and will have expensive cost if we annotate for a large number of novel classes. Therefore, our method has the unique advantage on transferring knowledge upon diffusion models to facilitate the unsupervised learning of multiple novel classes, which is irreplaceable against few-shot methods.

**Comparison with unsupervised setting.** We set representative unsupervised baselines including AutoLink [17] and Hedlin *et al.* [18] . Considering that existing unsupervised methods [17, 18] mostly focus on single-class scenario, we conduct the comparison using two splits. In the split "Novel Bird", the novel classes are all the bird classes in MP-100, where the unsupervised methods [17, 18] could be well applied. In the "MP-100 Split1", the novel classes contain multiple classes, *e.g.*, Bed and Lion. As shown in Tab. 3, the unsupervised methods Hedlin *et al.* [18] dramatically degrades when changing the split from "Novel Bird" to "MP-100 Split1", probably because that they require to pre-define a keypoint number and thus are not intuitive to tackle multiple classes having different keypoint numbers. Our method outperforms unsupervised method Hedlin *et al.* [18] dramatically in both splits, because our method could simultaneously tackle multiple novel classes and leverage the transferred knowledge.

## 4.5 Method Analysis

**Analysis on keyness transfer.** We learn keyness from labeled base classes and transfer to unlabeled novel classes, and thus the transferability of keyness is a key factor of our method. We quantitatively evaluate the estimated keyness map on base classes and novel classes to investigate its transferability. Tab. 4 summarizes the averaged L2↓ distances (after multiplying 100 as in [18] for better readability)

Table 2: Comparison with few-shot baselines.

| Method | Extra annotated Novel Classes | MP-100 Split1 | | MP-100 Split2 | |
|---|---|---|---|---|---|
| | | mAP↑ | L2↓ | mAP↑ | L2↓ |
| PoseAnything [19] | √ | 41.7 | 27.7 | 32.7 | 36.0 |
| MetaPoint [9] | √ | 42.3 | 27.2 | 33.1 | 35.8 |
| SCAPE [34] | √ | 42.7 | 27.1 | 33.8 | 34.4 |
| Ours | × | 70.4 | 18.4 | 61.3 | 26.2 |

Table 3: Comparison with unsupervised baselines.

| Method | Off-the-shelf Base Labels | Novel Bird | | MP-100 Split1 | |
|---|---|---|---|---|---|
| | | mAP↑ | L2↓ | mAP↑ | L2↓ |
| AutoLink [17] | × | 14.5 | 60.7 | 8.2 | 70.1 |
| Hedlin *et al.* [18] | × | 32.6 | 37.4 | 24.9 | 41.7 |
| Ours | √ | 56.1 | 22.1 | 70.4 | 18.4 |

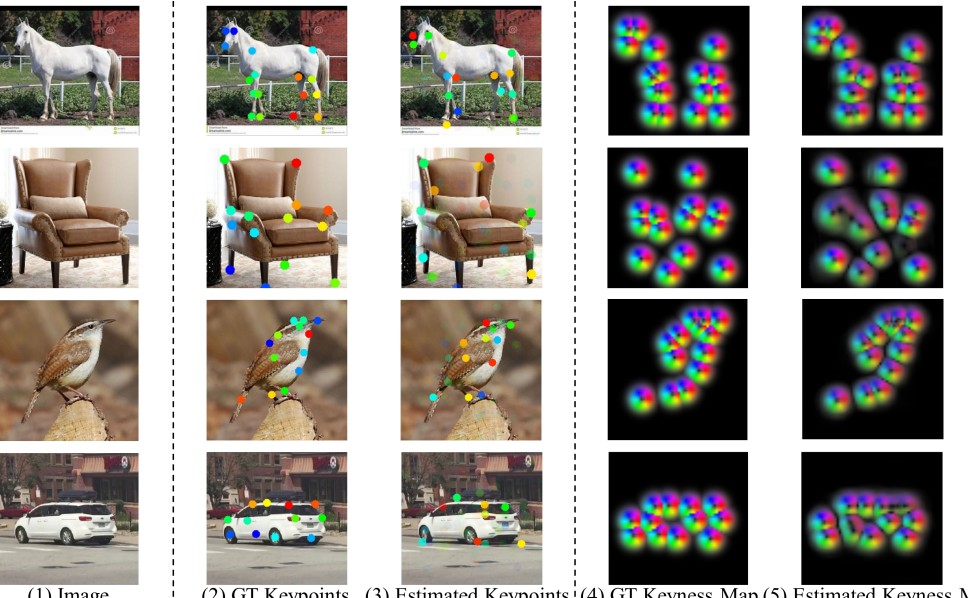

(1) Image    (2) GT Keypoints    (3) Estimated Keypoints    (4) GT Keyness Map    (5) Estimated Keyness Map

Figure 4: Qualitative analysis of estimated keypoints and keyness. The col (2,3) show the GT and estimated keypoints, while col (4,5) show the GT and estimated keyness. The keyness is shown by colors, where the hue indicates the direction angle and the saturation denotes the magnitude.

between estimated keyness maps and GT keyness maps for both base classes and novel classes on different splits. The averaged L2 distance before training is about $80$ (*i.e.*, the lower bound of performance), and we can see that the L2 distances of base classes are relatively lower because of the full supervision. The L2 distances of novel classes without full supervision satisfactorily approximate to the results of base classes, indicating that the keyness are class-wise transferrable.

To further explore the transferability of keyness, we visualize the estimated keypoints and keyness map in Fig. 4 in the intermediate training stage to see whether the keyness could provide beneficial guidance. Generally, the estimated keyness maps are more approximate to their GT than the estimated keypoints. Therefore, some biased keypoints (*e.g.*, the right eye of horse in the top row) could be guided to offset to the most key pixels, leading to precise keypoints of novel classes. Although some cases (*e.g.*, the sofa in the 2-nd row) are still imperfect, we are the first to explore unsupervised learning of multiple classes, which is quite challenging and has never been explored before. Overall, the transferred keyness maps are beneficial, and provide valid supervision to learn precise keypoints.

**Analysis on correspondence transfer.** The learned and transferred correspondences provide beneficial regularization for the unsupervised learning of novel classes. We evaluate their PCKs according to the GT keypoints on $x_i$ and the GT keypoints corresponded from $x_r$ to $x_i$. As shown in Tab. 4, the PCKs of base classes are high enough due to full supervision. And the PCKs of novel classes are relatively satisfactory, indicating that the transferred correspondences can provide valid regularization.

**Ablation study.** To study the contributions of our loss terms, we evaluate the combinations of our basic model, $\mathcal{L}_{NU}, \mathcal{L}_{NC}$ and $\mathcal{L}_{NG}$. As shown in Tab. 5, enabling the unsupervised loss $\mathcal{L}_{NU}$ could improve the mAP dramatically, indicating the primitively gains of using unannotated images. Solely enabling the consistency loss $\mathcal{L}_{NC}$ promotes the performance more significantly. Furthermore, incor-

Table 4: The performance of keyness (L2↓) and correspondence (PCKs↑) evaluated for base classes and novel classes on five splits.

| Method | Split | S1 | S2 | S3 | S4 | S5 |
|---|---|---|---|---|---|---|
| Keyness | Base | 47.2 | 43.6 | 45.3 | 42.2 | 44.8 |
| | Novel | 53.0 | 49.6 | 51.6 | 48.7 | 50.9 |
| Correspondence | Base | 89.1 | 94.0 | 93.3 | 94.8 | 93.5 |
| | Novel | 73.5 | 74.8 | 69.8 | 69.9 | 71.2 |

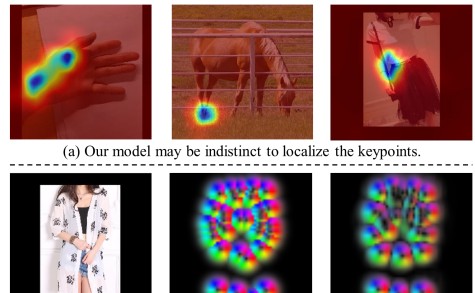

(a) Our model may be indistinct to localize the keypoints.

| (1) Image | (2) GT Keyness Map | (3) Estimated Keyness Map |

(b) Our model may fail to estimate fine-grained keyness.

Figure 5: The illustrations of two limitations.

Table 5: Ablation results of our proposed framework on two splits.

| Basic | $\mathcal{L}_{NU}$ | $\mathcal{L}_{NC}$ | $\mathcal{L}_{NG}$ | MP-100 Split1 mAP↑ | L2↓ | MP-100 Split2 mAP↑ | L2↓ |
|---|---|---|---|---|---|---|---|
| √ | | | | 60.2 | 22.4 | 51.5 | 29.0 |
| √ | √ | | | 65.8 | 20.1 | 56.7 | 27.5 |
| √ | | √ | | 66.5 | 20.2 | 57.6 | 27.9 |
| √ | | | √ | 67.7 | 19.5 | 58.2 | 27.3 |
| √ | √ | √ | √ | 70.4 | 18.4 | 61.3 | 26.2 |

Table 6: The statistical analysis of our method against the strongest baseline (*i.e.*, MetaPoint[9]). We summarize the "mean ± std" of mAP↑ and report the p-values against the baseline.

| Dataset Split | MetaPoint [9] | Ours | P-Value |
|---|---|---|---|
| S1 | 31.5±0.61 | 70.4±0.53 | 0.0 |
| S2 | 36.2±0.72 | 61.3±0.89 | 0.0 |
| S3 | 30.1±0.73 | 60.9±0.78 | 0.0 |
| S4 | 26.8±0.69 | 61.7±0.62 | 0.0 |
| S5 | 36.9±0.71 | 58.9±0.56 | 0.0 |

porating $\mathcal{L}_{NG}$ alone also demonstrates notable performance improvements. With all losses enabled, our method achieves the best results. Thus, our loss terms are effective and complementary.

### 4.6 Limitation Discussion

Our weak-shot keypoint estimation is prone to suffer from various issues due to insufficient annotations of multiple novel classes. We summarize three major problems found in practice. Firstly, our model is sometimes difficult to precisely localize keypoints via argmax on heatmaps, as shown in Fig. 5 (a). For example, in the first sub-figure, there are two maximums, and thus we may need better strategies to localize keypoints from heatmaps. Secondly, our estimated keyness map may fail to provide the directions to key directions, as shown in Fig. 5 (b). Our model is difficult to estimate precise or fine-grained keyness map in the chest area of the "long_sleeved_outwear", probably because the GT keypoints are too dense and crowded. Thirdly, we method may suffer from the imbalance between the image numbers of base classes and novel classes. Nevertheless, we are the first to explore such challenging cases, and we would like to address above limitations in future works.

### 4.7 Significance Test

In this section, we statistically analyse our framework against the most competitive baseline Meta-Point [9] on 5 splits of MP-100 dataset using 3 labeled images per base class and 30 unlabeled images per novel class. With random seeds ranging from 1 to 10, we run both methods for 10 times. Under the significance level 0.05, we conduct the significance test to show that our method is superior than MetaPoint [9]. We summarize the "mean ± std" of mAPs↑ results and p-values in various dataset splits in Tab. 6, where we could see that the p-values are all far below the significance level 0.05. Therefore, this statistical analysis demonstrates that the improvements of our proposed framework is statistically significant.

## 5 Conclusion

In this paper, we have explored weak-shot keypoint estimation, where multiple novel classes are learned from unlabeled images with the support of labeled base classes. We have proposed a novel framework transferring keyness and correspondence to facilitate the unsupervised learning of novel classes. By transferring keyness and correspondence, our framework has achieved favourable performance for weak-shot keypoint estimation. We have conducted comprehensive experiments on MP-100 dataset to demonstrate our effectiveness.

## Acknowledgements

This work was supported in part by the National Key Research and Development Program of China under Grant 2023YFE0210700, in part by the National Natural Science Foundation of China under Grants 62441203, 62311530101, 62132006, 62471287, 62402201, 62161013, in part by the Natural Science Foundation of Jiangxi Province of China under Grants 20252BAC240197, in part by the China Postdoctoral Science Foundation under Grant 2025M771495, and in part by the Early-Career Young Scientists and Technologists Project of Jiangxi Province under Grant 20244BCE52070.

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
