# OpenReview forum: "Weak-shot Keypoint Estimation via Keyness and Correspondence Transfer"
_NeurIPS.cc/2025/Conference — NeurIPS 2025 poster_

### Official Review · Reviewer_4nj5 · 2025-06-21

**Clarity:** 2
**Significance:** 3
**Originality:** 2
**Rating:** 4
**Confidence:** 3

**Summary:**

This paper introduces the concept of weak-shot learning for pose estimation, wherein the model is trained using labeled examples from base classes and unlabeled examples from novel classes. The proposed method processes pairs of images belonging to the same class, encoding them through a shared encoder and a latent diffusion model to estimate keypoints. Additionally, the approach generates keyness maps, indicating the direction toward the next keypoint, and establish correspondences between the images in each pair. For labeled images from base classes, a supervised loss is employed, while an unsupervised loss is proposed for unlabeled images from novel classes. The effectiveness of the method is demonstrated on various splits of the MP-100 dataset, with comparisons made against both unsupervised and few-shot learning approaches.

**Questions:**

Table 1 in the supplementary material indicates that shuffling the keypoint order significantly degrades the performance of Hedlin et al., despite this method being unsupervised. What factors might account for this observation?

How computationally expensive is the computation of the correspondence map, considering that the approach requires calculating the cosine similarity between every pair of pixels?

Is the "reference image" truly a reference in the conventional sense? According to my understanding, it simply refers to another image belonging to the same class. If so, this terminology may be misleading.In Equation 5, M_i* does not denote ground truth, whereas the same symbol is used to represent the ground-truth keyness map in Equation 4. This inconsistency could lead to reader confusion.

**Ethical Concerns:**

["NO or VERY MINOR ethics concerns only"]

**Final Justification:**

My concerns have been clarified, and I maintain the score.

**Paper Formatting Concerns:**

No concerns

**Quality:**

2

**Strengths And Weaknesses:**

Strengths:

The authors are first to apply the concept of weak shot learning on pose estimation, which is of high relevance due to the sparsity of labelled images in this field.

The method outperforms existing unsupervised and few-shot approaches by a large margin under the given data availability. The method even outperforms few-shot methods that have additional labels of novel classes.

The paper shows evidence that the addition of the keyness maps substantially enhances the performance in comparison to prior works.

Weaknesses:

Missing details on evaluation.

To enable fair comparisons between methods, it is necessary to specify the number of labeled/unlabeled training samples and classes used for each method.

Additionally, the authors should elaborate on the degree of similarity between the base and novel classes; while Appendix A4 provides some information, this discussion should be extended and included in the main text.

The overall clarity of the paper can be improved due to missing details and misleading terminology (see below).

---

> ### Author Rebuttal · Authors · 2025-07-31
>
> We sincerely thank the reviewer for the positive feedback and constructive suggestions. Below, we provide detailed responses to address each of the concerns raised.
>
> ---
>
> > **Q1:** To enable fair comparisons between methods, it is necessary to specify the number of labeled/unlabeled training samples and classes used for each method.
>
> **A1:** Below, we firstly specify the number of base/novel classes, and then specify the number of labeled/unlabeled training samples.
>
> The number of classes is determined by the dataset split.
> For the standard splits (i.e., MP-100 Split1-Split5), we follow [71,57,9] and employ 70/20 for base/novel classes, as introduced in Line 212-214. For the cross super-class splits in Table 4 of Appendix, we follow [71,57,9] and employ 99/1, 98/2, 95/5 and 97/3 for base/novel classes in Body, Face, Fur. and Veh. splits. For the Novel Bird split in Table 3, we employ 92/8 for base/novel classes.
>
> The numbers of samples and labels for base/novel classes are determined by the learning paradigm. Specifically, all methods in weak-shot learning (e.g., all methods in Table 1 and Ours in Table 2 and Table 3) employ 3/30 labeled/unlabeled training samples for each base/novel class, as introduced in Line 214-217. For the few-shot methods (with 1-shot) in Table 2, we employ 3/1 labeled/labeled training samples for each base/novel class. For the unsupervised methods in Table 3, we employ 30 unlabeled training samples for each novel class.
>
> Thanks again for the beneficial comments, and we will add the specifications in our revision.
>
> ---
>
> > **Q2:** Additionally, the authors should elaborate on the degree of similarity between the base and novel classes; while Appendix A4 provides some information, this discussion should be extended and included in the main text.
>
> **A2:** We measure the similarity via the GT keypoints and a ResNet-50 backbone pre-trained on ImageNet. Specifically, we firstly extract an image feature vector for each image, by averaging the feature vectors of GT keypoints on the backbone feature map. Then, we extract a class feature vector, by averaging the image feature vectors belonging to the same class. After that, for each novel class, we compute its similarity to base classes by averaging its similarities to all base classes using class feature vectors. Finally, we obtain the similarity from novel classes to base classes by averaging the similarities of all novel classes.
>
> In the table below, we summarize the similarities for all dataset splits. We could see that, the similarities of standard splits (i.e., Split1-Split5) are robust and intermediate due to random splitting. The similarities of cross super-class splits are significantly lower than the standard splits.
>
>
> |         | Split1 | Split2 | Split3 | Split4 | Split5 | Body   | Face   | Vehicle | Furniture |
> | ------------ | ------ | ------ | ------ | ------ | ------ | ------ | ------ | ------- | --------- |
> | **Cosine Similarity**  | 0.65 | 0.64 | 0.65 | 0.65 | 0.63 | 0.62| 0.61 | 0.54  |  0.59  |
>
>
> > **Q3:** Table 1 in the supplementary material indicates that shuffling the keypoint order significantly degrades the performance of Hedlin et al., despite this method being unsupervised. What factors might account for this observation?
>
> **A3:** Firstly, we adapt related methods (e.g., Hedlin et al. [21]) into weak-shot learning paradigm for fair comparison, as stated in Line 243-251. Secondly, our default learning paradigm is weak-shot learning, including the experiments in Table 1 of Supplementary. Therefore, in Table 1 of Supplementary, Hedlin et al. is an adapted weak-shot method, which significantly degrades due to overfitting unpractical relation, as explained in Line 218-223. The comparison to original unsupervised methods (e.g., Hedlin et al. [21]) is presented in Table 3 of main text, which are inferior due to the various keypoint numbers of multiple classes. Thanks again, and we will introduce distinguishing markers to differentiate the original method from its adapted version.
>
> ---
>
> > **Q4:** How computationally expensive is the computation of the correspondence map, considering that the approach requires calculating the cosine similarity between every pair of pixels?
>
> **A4:** Overall, we follow the implementation of [44] to compute correspondences. In practice, we only need to compute the similarity between all keypoints on image1 and all pixels on image2. In this way, the computation of such correspondence is relatively efficient and affordable, i.e., 0.66 ms and 0.37 GFLOPs. Specifically, the implementations with tensor shapes annotated are shown below:
> ```
>
> img1_feature_map = normalize_feats(img1_feature_map) # shape: (B, K, C)
> img2_feature_map = normalize_feats(img2_feature_map) # shape: (B, W*H, C)
> correspondence_map = torch.matmul(img1_feature_map, img2_feature_map.permute((0, 2, 1))) # shape: (B, K, W*H)
>
> def normalize_feats(feats):
>     feats = feats / torch.linalg.norm(feats, dim=-1)[:, :, None]
>     return feats
> ```
> Specifically, for the time cost, the spent time is 0.66 ms, which is averaged over 500 training iterations. For the theoretical cost, the GFLOPs is 0.37, which is measured by the fvcore package through aggregating all floating-point operations of feature normalization to unit vectors and matrix multiplication.
>
> Thanks again for the beneficial comments, and we will add more specifications in our revision.
>
> ---
>
> > **Q5:** Is the "reference image" truly a reference in the conventional sense? According to my understanding, it simply refers to another image belonging to the same class. If so, this terminology may be misleading. In Equation 5, $M_i^*$ does not denote ground truth, whereas the same symbol is used to represent the ground-truth keyness map in Equation 4. This inconsistency could lead to reader confusion.
>
> **A5:** Our "reference image" just means another image belonging to the same class, and we may employ "paired image" for better understanding. In Equation 5, we regard the transferred keyness map as pseudo ground truth, and we will remove the marker * for better consistency and readability.

---

> > ### Comment · Reviewer_4nj5 · 2025-08-07
> >
> > Thank you for your explanation and clarifications. My concerns have been clarified and I maintain the score.

---

> > > ### Author Response · Authors · 2025-08-08
> > >
> > > Thanks for the beneficial suggestions and positive rating. We will revise and improve our paper accordingly.

---

### Official Review · Reviewer_zjHs · 2025-06-26

**Clarity:** 3
**Significance:** 4
**Originality:** 3
**Rating:** 5
**Confidence:** 1

**Summary:**

This paper proposes a weak shot pose estimation, where multiple novel classes are learned from unlabeled images from the labeled base classes.

The proposed method leverages keyness (gradients of the keypoint that indicate the direction to a more important key location) and correspondence from the base class by transferring them to a novel class to form the unsupervised learning of the novel classes. Keyness and correspondences are estimated from the diffusion model. Novel classes are trained from offset guiding loss, consistency loss, and localization loss, while the base class is trained with keypoint loss, keyness loss, and correspondence loss.

The paper shows an experiment comparing the results on the baselines with similar setups as well as a fully supervised setup, and shows that the proposed method outperforms the baseline with the similar setup, while performs comparable results on the fully supervised setup.

**Questions:**

- Found one typo in line 130. I assume it's "single-class" instead of "sing-class"?
- Estimating keypoints is not necessarily a pose estimation. It would be great to avoid using the term "pose estimation" too many times in the paper.
- It would be great to have actual name on each loss term when they are introduced, for example in line 153: correspondence via XXX, XXX, XXX loss in base class (L_BP, L_BK, and L_BC), and line 154: keypoints via XXX, XXX, XXX loss in novel class (L_LG, L_NC, and L_NU)

**Ethical Concerns:**

["NO or VERY MINOR ethics concerns only"]

**Final Justification:**

Therer weren't any major reason to change the final rating

**Limitations:**

Yes

**Quality:**

3

**Strengths And Weaknesses:**

1. Strength:

- The paper is not my field, but the proposed method covers a less explored aspect of keypoint estimation, and technically sound & the experiment results look reasonable. I don't find any noticeable errors in the paper that need to be fixed. Also, the proposed method has great potential to be used in other computer vision fields.

2. Weakness:

- It's possible that this issue is coming from a terminology difference between different fields, but as a person from the 6D pose estimation field, the term "Pose Estimation" is somewhat misleading. A better term to be used is "Keypoint Estimation". The paper is mainly focused on keypoint estimation, and no where to find actual "pose" estimation.

---

> ### Author Rebuttal · Authors · 2025-07-31
>
> We sincerely thank the reviewer for the insightful comments and constructive feedback. Below, we address each concern and question raised.
>
> ---
>
> > **Q1:** It's possible that this issue is coming from a terminology difference between different fields, but as a person from the 6D pose estimation field, the term "Pose Estimation" is somewhat misleading. A better term to be used is "Keypoint Estimation". The paper is mainly focused on keypoint estimation, and no where to find actual "pose" estimation.
>
> **A1:** We intuitively follow the related works Category-Agnostic Pose Estimation [71,57,9] to use the term of "Pose Estimation", and we will revise it to "Keypoint Estimation" for better understanding.
>
> ---
>
>
> > **Q2:** Found one typo in line 130. I assume it's "single-class" instead of "sing-class"?
>
> **A2:**  Sincerely thanks for the meticulous comments, and we will correct the typo and revise the paper more carefully.
>
> ---
>
>
> > **Q3:** Estimating keypoints is not necessarily a pose estimation. It would be great to avoid using the term "pose estimation" too many times in the paper.
>
> **A3:** Thanks for the insightful comments, and we will employ the precise term, i.e., keypoint estimation.
>
> ---
>
>
> > **Q4:** It would be great to have actual name on each loss term when they are introduced, for example in line 153: correspondence via XXX, XXX, XXX loss in base class ($\mathcal{L}\_{BP}$, $\mathcal{L}\_{BK}$, and $\mathcal{L}\_{BC}$), and line 154: keypoints via XXX, XXX, XXX loss in novel class ($\mathcal{L}\_{LG}$, $\mathcal{L}\_{NC}$, and $\mathcal{L}_{NU}$)
>
> **A4:** Thanks for the beneficial comments, and we will add the actual name in the descriptions to improve the readability.

---

> ### Comment · Reviewer_zjHs · 2025-08-08
>
> I do not find any significant reason to change my rating. Thanks to the authors for the response.

---

> > ### Author Response · Authors · 2025-08-09
> >
> > Thank you for providing the valuable suggestions and for the positive rating. We will revise and improve our paper accordingly.

---

### Official Review · Reviewer_YayZ · 2025-06-29

**Clarity:** 3
**Significance:** 3
**Originality:** 3
**Rating:** 4
**Confidence:** 3

**Summary:**

The paper proposes a weak-shot keypoint detection approach, which aims to detect semantic-rich keypoints on new classes without labeled training data.
The method is first trained on labeled data and then reuses the pretrained prompt tokens to detect keyness maps on new classes. Then, in new classes, the keyness map is used to guide the pixel movement to high-intensity change regions, where the semantic correspondences across images are used to provide consistency loss. Experiments are conducted on the MP-100 dataset.

**Questions:**

(1) Can authors provide a clearer definition of "new class"?

(2) Given that the prompt tokens are pretrained exclusively on the base categories, could using them during the unlabeled training phase for novel categories inadvertently introduce bias?

**Ethical Concerns:**

["NO or VERY MINOR ethics concerns only"]

**Final Justification:**

After reading other reviewers' comments and discussions, the reviewer decided to maintain the original positive score.

**Limitations:**

Yes

**Quality:**

3

**Strengths And Weaknesses:**

# Strengths
(1) The proposed keyness constraint and correspondence constraint on the new class are reasonable and seem effective

(2) Experiments and analysis are well conducted. The paper provides comprehensive ablation studies to validate the effectiveness of the proposed method.

# Weaknesses

(1) Definition of "class". It is not clear to me whether new class means new category (cat to dog) or new instance (cat 1 to cat 2). The paper seems to focus on a new instance. Otherwise, the reuse of the pretrained prompt tokens on different categories may introduce severe bias towards the new category.

(2) Task definition. The paper seems to estimate a set of keypoints that can be matched across different instances. However, the reviewer thinks this work may not belong to the pose estimation task, which aims to detect the predefined set of semantic keypoints, and can benefit the downstream tasks like action recognition, skeleton-driven animation, etc.
If the reviewer does not misunderstand, the detected keypoints on the new class in this work are only "semantic-rich", but don't have specific (known) semantic meaning.
Therefore, it should be a semantic keypoint detection work instead of a pose estimation work.

---

> ### Author Rebuttal · Authors · 2025-07-31
>
> We sincerely thank the reviewer for the positive evaluation of our work and for acknowledging the insights of our method. Below, we provide detailed responses to address each of the concerns raised.
>
> ---
>
> > **Q1:** Definition of "class". It is not clear to me whether new class means new category (cat to dog) or new instance (cat 1 to cat 2). The paper seems to focus on a new instance. Otherwise, the reuse of the pretrained prompt tokens on different categories may introduce severe bias towards the new category.
>
> > Can authors provide a clearer definition of "new class"?
>
> **A1:** We will add explicit definition, i.e., new class means new category, which follows the definition in related works [71,57,9]. Exactly, the cross-category bias is one of the major issues to address in our task. We will further explain our contributions to this issue in the following Q3-A3.
>
> ---
>
>
> > **Q2:** Task definition. The paper seems to estimate a set of keypoints that can be matched across different instances. However, the reviewer thinks this work may not belong to the pose estimation task, which aims to detect the predefined set of semantic keypoints, and can benefit the downstream tasks like action recognition, skeleton-driven animation, etc. If the reviewer does not misunderstand, the detected keypoints on the new class in this work are only "semantic-rich", but don't have specific (known) semantic meaning. Therefore, it should be a semantic keypoint detection work instead of a pose estimation work.
>
>
> **A2:** We intuitively follow the related works Category-Agnostic Pose Estimation [71,57,9] to use the term of "Pose Estimation", and we will revise our task name to more precise one, i.e., Keypoint Detection or Keypoint Estimation.
>
> As for the downstream tasks, our estimated keypoints have similar properties as the ones in unsupervised keypoint detection [21,19,83], which can also benefit the downstream tasks such as action recognition. In particular, optical flow is semantic-free, but is beneficial for action recognition. Our estimated keypoints may be regarded as sparse optical flow, and the keypoint trajectories across video frames could depict the motion pattern. Besides, as in the regression-based evaluation [21,19,83] described in Line 237-241, the estimated keypoints can be regressed to known semantic meaning.
>
> Discovering keypoints with known semantic meaning is an interesting yet challenging problem, and we will explore more in future works. Our primary focus in this paper is on discovering keypoints for multiple classes, which also has received relatively limited exploration.
>
> ---
>
>
> > **Q3:** Given that the prompt tokens are pretrained exclusively on the base categories, could using them during the unlabeled training phase for novel categories inadvertently introduce bias?
>
> **A3:** Indeed, and addressing this cross-category domain gap (bias) is one of the major issues. Preliminarily, reusing the prompt tokens for novel categories is effective due to the keypoint sharing between base categories and novel categories. For example, "bird" and "lion" both have eye keypoints. Moreover, we propose to employ transferred keyness and correspondence to further bridge the domain gap (bias). As explained in Line 41-46, both keyness and correspondence belong to comparing entities(or relations), which are highly transferable. By above designs, our model outperforms existing methods dramatically in learning keypoints for multiple classes without labels.

---

> > ### Comment · Reviewer_YayZ · 2025-08-07
> >
> > The reviewer appreciates the authors' efforts in rebuttal and has carefully read other reviewers' comments. The reviewer has no more questions at this time.

---

> > > ### Author Response · Authors · 2025-08-08
> > >
> > > Thanks for the insightful comments and positive rating. We will revise and enhance our paper accordingly.

---

### Official Review · Reviewer_iFGi · 2025-07-03

**Clarity:** 2
**Significance:** 3
**Originality:** 3
**Rating:** 4
**Confidence:** 2

**Summary:**

This paper proposes a new task, named weak-shot pose estimation, and designs a framework to transfer keyness and correspondence from base classes to novel classes for the focused task. To address the lack of annotations for novel classes,  an offset guiding loss and a consistency loss are introduced to guide the learning of heatmap, keyness, and correspondence. Experiments are conducted on MP-100 benchmark to demonstrate the effectiveness of the proposed method.

**Questions:**

- What's the meaning of K In Eq. (3) and Eq. (5)? Why not M? Is K the exact number of keypoints?

- No illustrations of $S$ and $w_{l,s}$ in Line141.

- In Table 5, does the “Basic” result in the first row refer to unsupervised setting? It still outperforms other methods under the weak-shot setting by a large margin. If so, what advantages does the proposed framework have over existing approaches?

- Typo: Line133 "\Phi_^c_l () and \Phi_^c_l () $ as the c-th ..."; Line183 "detach and block" -> "detaches and blocks"

**Ethical Concerns:**

["NO or VERY MINOR ethics concerns only"]

**Limitations:**

Yes.

**Paper Formatting Concerns:**

None.

**Quality:**

3

**Strengths And Weaknesses:**

##  Strength

- The paper proposes the new task of weak-shot pose estimation, and designs a framework to transfer keyness and correspondence from base classes to novel classes.

- Self-supervised losses on novel classes, e.g., offset guiding loss and consistency loss, are introduced to guide the learning of  heatmap, keyness, and correspondence.

- The method significantly outperforms the established baselines by a large margin on MP-100 benchmark.

- The paper is well-organized.


## Weakness

- How is the keyness map $M$ used to identify the corresponding keypoint for each pixel?

- For the base classes, how is the visibilities $V$ supervised?

- The description of the offset-guiding loss (Lines 179–184) is somewhat unclear.  How is the visibilites $V$ obtained via the argmax operation? Is there any thresholding involved? How exactly is Eq. (5) used to supervise visibility? Does $M_i^*[P_i[k]]$ refer to the keyness at pixel $P_i[k]$? What is the precise operation of dt()? How does the gradient propagate through $P$ given the use of argmax operations?

---

> ### Author Rebuttal · Authors · 2025-07-31
>
> We sincerely appreciate the reviewer’s positive feedback and constructive suggestions. Below, we provide point-by-point responses to address each concern raised.
>
> ---
>
> > **Q1:** How is the keyness map $M$ used to identify the corresponding keypoint for each pixel?
>
> **A1:**  Actually, the keyness map $M\in \mathbb{R}^{2\times H\times W}$ presents the directions towards more key positions, and we employ it to guide the estimated keypoints to move to more key coordinates. For example, given coordinates $p=[x,y]$, the 2D offset vector $M[p]=[\Delta x,\Delta y]$ fetched from the keyness map $M$ indicates the more key direction. Therefore, the primary role of keyness map $M$ is to guide the keypoint coordinates, rather than identifying the corresponding keypoint for each pixel.
>
> ---
>
> > **Q2:** For the base classes, how is the visibilities $V$ supervised?
>
>
> **A2:** In our model, the visibilities $V$ are the max values fetched from heatmaps (Line 179-180), and thus are indirectly supervised by the loss on heatmaps (i.e., $\mathcal{L}_{BP}$ at Eq. (3)). For the base classes, the matched heatmaps are supervised to approaching the GT keypoint Gaussian maps (max=visibility=1), while the unmatched heatmaps are supervised to approaching all-zero maps (max=visibility=0). Considering that the supervision on unmatched estimations is trivial and similar to [6,11,9], we omit it for brevity. Thanks for the considerate comments, and we will add more explanations in our revision.
>
> ---
>
> > **Q3:** The description of the offset-guiding loss (Lines 179–184) is somewhat unclear. ① How is the visibilites $V$ obtained via the argmax operation? ② Is there any thresholding involved? ③How exactly is Eq. (5) used to supervise visibility? ④Does $M_i^*[P_i[k]]$ refer to the keyness at pixel $P_i[k]$? ⑤What is the precise operation of dt()? ⑥How does the gradient propagate through $P$ given the use of argmax operations?
>
> **A3:** **The answers to ①②④⑥**: Actually, we employ "soft" argmax operation to obtain coordinates and employ max operation to obtain visibilities, as stated in Line 179-180.
>
> Specifically, given one keypoint heatmap $H$, we compute the keypoint coordinates $\tilde{\mathbf{p}}\in \mathbb{R} ^{2}$ by following "soft" argmax: $\tilde{\mathbf{p}}= \sum_{\mathbf{p}} w(\mathbf{p}) \mathbf{p}$, where   $w(\mathbf{p}) = \frac{\exp(H[\mathbf{p}])}{\sum_{\mathbf{p}} \exp(H[\mathbf{p}])}.$ Here, the index $\mathbf{p}$ denotes the normalized pixel coordinates (all spatial grid points), and $H[\mathbf{p}]$ denotes fetching the value at $\mathbf{p}$ from $H$. Similarly, $M_i^*[P_i[k]]$ refers to the keyness at pixel $P_i[k]$.
>
> For keypoint visibility, we compute by $v_n = \underset{\mathbf{p}}{\mathrm{max}} H[\mathbf{p}]$. We do not involve thresholding for computing visibility, because soft values are compatible for supervision and evaluation. The above "soft" argmax operation is differentiable, and identical operations could be found in [18,57].
>
> **The answers to ③**: As also stated in our Abstract and Introduction, the primary role of Eq. (5) is providing direct guidance for keypoint coordinates. We sincerely apologize for the rigor of expression of "and visibilities" at Line 181, and we will remove it in the final version.
>
>
> **The answer to ⑤**: The precise operation of dt() is torch.detach().
>
> ---
>
> > **Q4:** What's the meaning of $K$ In Eq. (3) and Eq. (5)? Why not $M$? Is $K$ the exact number of keypoints?
>
> **A4:** As described in Line 118-120, $K$ is the exact number of keypoints (including padded and invisible keypoints). Besides, we do not employ $M$ to represent a certain number, but employ $N$ to represent the number of keypoint prompts and thus estimate $N$ keypoint heatmaps for each image.
>
> The loss term in Eq. (3) supervises on $K$ matched heatmaps, because there are $K$ GT keypoints for matching. As for Eq. (5), the loss term should average over $N$, i.e., $\frac{1}{N}\sum_k^N$, because the guidance is applicable for all estimated coordinates, which can also be inferred from the dimensional notation $N$ at Line 180. We sincerely apologize for this symbol mistake, and will revise in the final version.
>
> ---
>
>
> > **Q5:** No illustrations of $S$ and $w_{l,s}$ in Line141.
>
> **A5:** Specifically, $S$ denotes the number of subsampled timesteps selected from the diffusion timesteps, and $w_{l,s}$ denotes the mixing weights for the $l$-th layer at the $s$-th timestep. We will add the illustrations in our final revision.
>
>
> ---
>
> > **Q6:** In Table 5, does the “Basic” result in the first row refer to unsupervised setting? It still outperforms other methods under the weak-shot setting by a large margin. If so, what advantages does the proposed framework have over existing approaches?
>
> **A6:** In the ablation study presented in Table 5, the default learning paradigm is weak-shot learning, with various loss terms activated. Thus, our basic model (the first row in Table 5) is not in unsupervised setting.
> Specifically, our basic model disables the loss terms for novel classes (i.e., $\mathcal{L}\_{NU}$,$\mathcal{L}\_{NC}$,$\mathcal{L}\_{NG}$), and learns from base classes and directly estimates for novel classes. Existing unsupervised approaches are designed for single-class scenario, and thus cannot: (1) well tackle different keypoint numbers of multiple classes; (2) well transfer across classes. Our basic model is trained with matching-based loss ( $\mathcal{L}\_{BP}$ at Eq. (3)). In this way, different number of keypoints could adaptively and dynamically match to keypoint prompts, and thus our basic model can well tackle multiple classes. Besides, the keypoints of novel classes and base classes may share the same keypoint prompts, and thus our prompts learned from base classes can also estimate precise keypoints for novel classes. Other loss terms (i.e., $\mathcal{L}\_{NU}$,$\mathcal{L}\_{NC}$,$\mathcal{L}\_{NG}$) can further improve.
>
> ---
>
>
> > **Q7:** Typo: Line133 "\Phi_^c_l () and \Phi_^c_l () $ as the c-th ..."; Line183 "detach and block" -> "detaches and blocks"
>
> **A7:** Sincerely thanks for the considerate comments, and we will correct the typos and revise the paper more carefully.

---

> > ### Comment · Reviewer_iFGi · 2025-08-06
> >
> > Thank you to the authors for responding to my concerns, most of which have been well addressed.
> >
> > I have a few additional minor questions:
> >
> > 1) Since keyness and correspondence are only used during training, are the final keypoints at test time obtained via an argmax over the heatmap H?
> >
> > 2) Why not use a keyness map 𝑀 with dimensions H*W*N, similar to the heatmap 𝐻? Would this help reduce ambiguities when keypoints are close to each other?

---

> > > ### Author Response · Authors · 2025-08-07
> > >
> > > Thanks for the further comments. We are pleased that most of your concerns were well addressed in our initial rebuttal. Below, we provide responses to the additional minor questions.
> > >
> > > ---
> > >
> > > __1.__ Yes. Therefore, the pipeline of our model in the test stage is consistent with Hedlin et al. [21].
> > >
> > > ---
> > >
> > > __2.__ Thanks for this interesting and insightful question. The desired keyness map ought to be class-agnostic and class-wise transferrable. Our single-channel keyness map is generated by comparing pixels in a general manner, making it class-agnostic. However, it may not be immediately intuitive to expand to N-channel while preserving the class-agnostic characteristic. We would like to explore more technical designs for the N-channel keyness to reduce ambiguities.

---

> > > > ### Comment · Reviewer_iFGi · 2025-08-07
> > > >
> > > > Thank you for the additional response. I am sorry for the confusion in my second follow-up question, where the intended dimension of M should be 2*H*W*N, rather than H*W*N, as it represents 2D offsets pointing to the most relevant pixels. This line of exploration might also involve leveraging the proposed prompts E, which could further enhance the learning process of  E.
> > > >
> > > > Overall, I would still maintain my positive rating.

---

> > > > > ### Author Response · Authors · 2025-08-07
> > > > >
> > > > > We appreciate the constructive suggestion and positive rating. We may employ the proposed prompts E or additional query embeddings to learn multiple and separate keyness maps, which could be beneficial for separating adjacent keypoints. We will explore various potentials on this line. Thanks again.

---

### Decision · Program_Chairs · 2025-09-17

**Decision:**

Accept (poster)

**Comment:**

This paper proposes a weak-shot pose estimation framework that transfers knowledge from base classes to novel classes through keyness and correspondence. The method demonstrates innovation, and the experiments and analyses are thorough, showing improvements over SOTA approaches.

​​Strengths:​​
1. The method is innovative and addresses a practically relevant problem.
2. The experimental results and analyses are comprehensive, validating the effectiveness of the approach.

​​Weaknesses:​​
1. As noted by the reviewers, further elaboration is needed on certain technical details, conceptual definitions, and experimental specifics.
2. Limitations and failure cases could be included in the main text to enhance structural completeness.

During the rebuttal phase, reviewers acknowledged the method's novelty and experimental validity, with discussions primarily focusing on clarifications regarding technical details and experiments (Reviewers iFGi, 4nj5, zjHs). Additionally, Reviewers YayZ and zjHs suggested refining the terminology, particularly replacing "pose estimation" with "keypoint estimation." The authors provided detailed responses in the rebuttal, committing to supplementing explanations in the paper, including technical details and experiments, as well as considering the terminology adjustment. The reviewers expressed satisfaction with the rebuttal, with final recommendations consisting of 3 ​​Borderline Accept​​ and 1 ​​Accept​​. The AC concurred with the reviewers' suggestions.